# Flexural Performance of RC Beams Strengthened with Externally-Side Bonded Reinforcement (E-SBR) Technique Using CFRP Composites

**DOI:** 10.3390/ma14112809

**Published:** 2021-05-25

**Authors:** Md. Akter Hosen, Fadi Althoey, Mohd Zamin Jumaat, U. Johnson Alengaram, N. H. Ramli Sulong

**Affiliations:** 1Department of Civil and Environmental Engineering, College of Engineering, Dhofar University, Salalah 2509, Oman; 2Department of Civil Engineering, College of Engineering, Najran University, Najran 1988, Saudi Arabia; 3Department of Civil Engineering, Faculty of Engineering, University of Malaya, Kuala Lumpur 50603, Malaysia; zamin@um.edu.my (M.Z.J.); johnson@um.edu.my (U.J.A.); hafizah_ramli@um.edu.my (N.H.R.S.); 4School of Civil & Environmental Engineering, Science and Engineering Faculty, Queensland University of Technology (QUT) 2 George St, Brisbane City, QLD 4000, Australia

**Keywords:** flexural strengthening, E-SBR, CFRP composites, ductility, energy absorption capability, stiffness

## Abstract

Reinforced concrete (RC) structures necessitate strengthening for various reasons. These include ageing, deterioration of materials due to environmental effects, trivial initial design and construction, deficiency of maintenance, the advancement of design loads, and functional changes. RC structures strengthening with the carbon fiber reinforced polymer (CFRP) has been used extensively during the last few decades due to their advantages over steel reinforcement. This paper introduces an experimental approach for flexural strengthening of RC beams with Externally-Side Bonded Reinforcement (E-SBR) using CFRP fabrics. The experimental program comprises eight full-scale RC beams tested under a four-point flexural test up to failure. The parameters investigated include the main tensile steel reinforcing ratio and the width of CFRP fabrics. The experimental outcomes show that an increase in the tensile reinforcement ratio and width of the CFRP laminates enhanced the first cracking and ultimate load-bearing capacities of the strengthened beams up to 141 and 174%, respectively, compared to the control beam. The strengthened RC beams exhibited superior energy absorption capacity, stiffness, and ductile response. The comparison of the experimental and predicted values shows that these two are in good agreement.

## 1. Introduction

Strengthening or rehabilitation of civil engineering infrastructures is gaining significant consequence due to the increasing deteriorated structures in numerous places all over the world. The common technique or method for flexural strengthening of RC members is externally bonded reinforcement (EBR) [1,2,3,4]. This strengthening technique is becoming more popular due to the prominent properties of CFRP such as high stiffness, strength to weight ratio, tremendous corrosion safeguard, non-conductivity, and ease of installation [5,6,7]. Over a few decades, widespread research has been conducted on the RC beams strengthening by EBR technique with CFRP sheets. Those sheets prominently improved the flexural responses of the beam specimens. The improvement depends on the numerous factors such as thickness, width, bonding length, and the number of layers of the CFRP sheet [8,9,10].

Ross et al. [11] carried out the study on the experimental and numerical characteristics of the RC beams strengthened with FRP sheets. The experimental results exhibited the premature failure of the FRP. Tumialan et al. [12] assessed experimental behavior on the RC beams strengthened by EBR CFRP laminate with the different number of plies and shear span under three-point bending. The tested beam specimen’s mode of failure shown the CFRP laminate or concrete delamination at the end of the laminate.

Leung, C.K.Y. [13] studied RC beam specimens strengthened with FRP plates by EBR technique. The outcomes of the tested beams show the different trend of debonding, which, depending on the thickness and size of plates and U-FRP stirrups, delay the debonding.

Toutanji, H., et al. [14] presented experimental and analytical studies of RC beam strengthened by EBR-FRP sheets using inorganic epoxy resin under flexural loads. Most of the beams failed by delamination of FRP sheets.

Therefore, the EBR method is not reasonable for RC beams strengthened for flexure due to the premature failure modes. Hence, the Externally-Side Bonded Reinforcement (E-SBR) technique is proposed to mitigate the limitations of the existing EBR method.

Insufficient studies have been found based on the side bonded method. Hosen et al. [15] studied the flexural improvement of RC beams by side near-surface mounted (SNSM) technique and existing near-surface mounted (NSM) method using glass fiber reinforced polymer bars. The experimental outcomes demonstrated that the SNSM technique had better flexural and energy dissipation capacities, stiffness, and ductility over the existing NSM method. The SNSM technique effectively eliminated the premature failure modes whereas the NSM strengthened specimens failed by debonding.

In this research, a new strengthening method, namely, Externally-Side Bonded Reinforcement (E-SBR) technique was applied. An experimental and analytical study of the flexural strengthened RC beams with the E-SBR technique was conducted utilizing CFRP fabrics. The two series of full-size RC beams were tested with different tensile steel reinforcing ratios. In total, six beams strengthened with CFRP fabrics by E-SBR technique with various width of side fabrics without and with U-anchorage, and two control beams, which were tested under the four-point loading. The load-deflection characteristics, strain data at varies positions and modes of failure of the beams were also captured. The strengthened beams results were compared with the control beam results. Analytical models based on the section analysis and strains compatibility were used to predict the load-deflection behavior, flexural load-bearing capacity, and average spacing and width of cracks of the beams. The analytical results were compared with the experimental findings.

## 2. Advantage of E-SBR Strengthening Technique

The main advantage of the E-SBR strengthening technique is easy to install the strengthening reinforcement at the lower portion on the side face of the existing RC beams for enhancing the flexural performance with improved ductility and the stiffness of beams in order to decrease the deflection and crack widths; further, it could also enhance the shear strength. Additionally, it is easier to increase the surface area for strengthening reinforcement when there is the demand to enhance the flexural capacity of the existing RC beam. The bonding between the concrete substrate and strengthening reinforcement can be improved by E-SBR technique.

## 3. Experimental Investigation

### 3.1. Test Matrix

In this study, the flexural behavior was investigated for RC beams strengthened with the E-SBR technique utilizing CFRP fabrics. A total of eight full-size beams were prepared for experimental investigation. The beams were categorized into two groups (A and B) based on the main reinforcing steel ratios. The two beams were control (CB-A and CB-B), two beams were strengthened with 42 mm width of E-SBR (SE-A and SE-B), two beams were strengthened with 84 mm width of E-SBR (SE-2A and SE-2B), and two beams were strengthened with 125 mm width of E-SBR and 100 mm width of U-wrap end anchorage (SE-3A and SE-3B). Table 1 shows the test matrix of the experimental program.

### 3.2. Test Specimens

The full size reinforced concrete beams with dimensions of 150 width × 250 height × 3300 total length mm^3^ and 3000 mm effective span length, as exhibited in Figure 1, was used in testing. The ratio between the shear span to the effective depth of the beams was kept as 5.11. The first and second series of beams were reinforced with two numbers of 10 mm and 12 mm diameter of deformed steel rebars, respectively, as a tensioned reinforcement. The hanger bars used for both series of specimens were two 10 mm diameter of deformed steel rebars until the shear span region, which was located at the top of the beams. The stirrups consist of 6 mm diameter round mild steel bars, which were symmetrically placed at the spacing of 75 mm for both series of the specimens. The detailing of the reinforcements for all specimens is revealed in Figure 1.

### 3.3. Material Properties

#### 3.3.1. Concrete

The beams were prepared by casting of ordinary mix concrete, which consists of ordinary Portland cement (OPC), 4.75 mm maximum size of fine aggregate (FA), and 20 mm maximum size of coarse aggregate (CA). The mixture of concrete was designed in accordance with the Department of Environment’s (DOE) method. Table 2 was contained the mix proportion of the concrete.

The compressive strength tests of concrete on 100 mm × 100 mm × 100 mm concrete cube specimens was conducted based on the BS EN 12390-3 [16], and the average compressive strength of the tested cubes for each beam specimens was found as 63.15 MPa. The modulus of rupture test using 100 × 100 × 500 mm^3^ concrete prism specimens were carried out following BS EN 12390-5 [17], and the average modulus of rupture of 5.78 MPa was recorded. The splitting tensile test was performed as per BS EN 12390-6 [18] using the concrete cylinder of 100 mm diameter and 200 mm height produced the average splitting tensile strength of about 4.52 MPa. The final test under mechanical properties of concrete was the modulus of elasticity test, carried out using 150 mm diameter and 300 mm height of concrete cylinders following the ASTM C469/C469M-14 [19], and the average modulus of elasticity of 36.55 GPa was found.

#### 3.3.2. Steel Bars

The beams casing was prepared using three types of deformed steel reinforcing rebars, as internal reinforcement. The high strength reinforcing rebars of 12 mm *ϕ* and 10 mm *ϕ* with yield strengths of 550 MPa and 520 MPa, respectively, were utilized as main reinforcements. The round mild steel reinforcing bars of 6 mm *ϕ* grade of 300 was used as a stirrup. The Young’s modulus was 200 GPa for all steel reinforcing steel bars.

#### 3.3.3. Adhesive

Sikadur^®^ 330 epoxy adhesive was applied as a structural bonding material among the strengthening reinforcement (CFRP fabrics) and the concrete surface of the beams. The epoxy adhesive comprised of part A (white color) and part B (grey color). Those two parts of adhesive were mixed by weight at a ratio of 4(A):1(B) to attain a uniform grey color. The density, compressive, tensile, and bond strength, and tensile modulus of elasticity of the epoxy as given by the supplier were 1.30 kg/l, 77.2 MPa, 30 MPa, 4 MPa, and 4.5 GPa, respectively [20].

#### 3.3.4. CFRP Fabrics

SikaWrap^®^-300C carbon fiber reinforced polymer (CFRP) unidirectional fabrics were applied for strengthening of RC beam specimens. The thickness, density, elongation at break, ultimate strength, and modulus of elasticity of the fabrics were 0.17 mm, 1.79 g/m^3^, 1.5%, 3.9 GPa, and 230 GPa, respectively.

### 3.4. Strengthening Procedure

The strengthening of RC beams in flexural applying CFRP fabrics was conducted by externally-side bonded reinforced (E-SBR) technique. The reinforced concrete beam side based on the E-SBR technique was prepared according to the instructions of the Sikadur-330 epoxy adhesive manufacturer. Primarily, a hand grinding appliance was exercised to remove the slack materials from the beam surface which might influence the bonding strength among the concrete substrate and CFRP fabrics, and attained a smooth beam surface. Secondly, a cleaning wear brush and high force dust blow air jet gun were applied to dirt-free the beam surface. Finally, acetone was utilized to eliminate any probable dust from the beam surface.

A tinny coating of epoxy adhesive resin was the blowout over the required beam surface to seal the trivial cavities and then CFRP fabrics was placed on it. The special bubble buster roller was steadily pressed on the CFRP fabrics until the epoxy resin flow out through the weeny openings of the unidirectional fabrics and removed additional epoxy among the concrete and fabrics. The E-SBR strengthened beams not permitted to move at least 72 h to attain the full bonding strength between epoxy and beam surface.

Two E-SBR strengthened RC beam specimens (SE-3A and SE-3B) were fastened at the end with 100 mm width of U-wrap by CFRP fabrics, as shown in Figure 2, to get rid of the premature peeling off (CFRP fabrics) failure of the specimens.

### 3.5. Test Setup

The experimental set-up and instrumentation were executed for successfully performing the tests of the beams in the heavy structure laboratory (Figure 3). The load and deflection at mid-span of the tested beams were taken from the data acquisition system of the Universal Instron machine. The tensile strain of steel reinforcements was measured by fastening a 30 mm strain gauges at the middle of the rebars. The compressive strain of concrete was measured by fixing a 30 mm strain gauge on the upper extreme fiber of concrete. The strain data of the specimens were stored in TDS-530 data logger throughout the experiments. The control and strengthened beams were tested below the four-point static loads. The shear span length of the specimens was 1150 mm. The tests were conducted by applying displacement control and the Instron machine actuator rate was 1.50 mm/min. The crack width of the beams was measured by Dino-Lite digital microscope throughout the test.

## 4. Analytical Model

The analytical models for the flexural load-carrying capacity of E-SBR-CFRP fabrics strengthened RC beams are developed applying the sectional analysis and strain distributions [21]. The models can predict the yield and ultimate loads, flexural crack spacing and width, and deflection of the control and strengthened beams.

### 4.1. Prediction of Load

The flexural load-bearing capacity of an E-SBR strengthened beam is computed from the compatibility of strain and equilibrium of forces as introduced in Figure 4.

#### 4.1.1. Yield Load

The yielding load (*P_y_*) of the E-SBR strengthened beams was calculated by considering the elastic limit of steel rebars. The yield load is calculated based on Equations (1)–(7).
(1)$$y=\frac{-(nA_{s}+n_{se}A_{se})+\sqrt{{\left(nA_{s}+n_{se}A_{se}\right)}^{2}+2b(nA_{s}d+n_{se}A_{se}d_{se})}}{b}$$
(2)$$I_{cr}=\frac{by^{3}}{3}+nA_{s}{\left(d-y\right)}^{2}+n_{se}A_{se}{\left(d_{se}-y\right)}^{2}$$
(3)$$\varepsilon_{y}=\frac{f_{y}}{E_{s}}$$
(4)$$\phi_{y}=\frac{\varepsilon_{y}}{d-y}$$
(5)$$M_{y}=\phi_{y}I_{cr}E_{c}$$
(6)$$E_{c}=4700\sqrt{{f^{\prime }}_{c}}$$
(7)$$P_{y}=\frac{2M_{y}}{L_{a}}$$

#### 4.1.2. Ultimate Load

The computation of ultimate load is based on the assumption that the failure occurred due to crushing of the extreme concrete fiber, followed by the rupture of the CFRP fabrics of the strengthened beams. The computation is executed in the constant region of the bending moment, which confirmed that no shear force is in this region. The ultimate load is calculated based on Equations (8)–(12).
(8)$$y=\frac{-(\varepsilon_{se}E_{se}A_{se}-A_{s}f_{y})+\sqrt{{\left(\varepsilon_{se}E_{se}A_{se}+A_{s}f_{y}\right)}^{2}+3.2b{f^{\prime }}_{c}(\varepsilon_{c}E_{se}A_{se}d_{se})}}{1.6b{f^{\prime }}_{c}}$$
(9)$$\varepsilon_{se}=\varepsilon_{c}\frac{d_{se}-y}{y}\leq \varepsilon_{seu}$$
(10)$$\varepsilon_{seu}=\frac{\sigma_{se}}{E_{se}}$$
(11)$$M_{u}=A_{s}f_{y}(d-0.4y)+\varepsilon_{c}E_{se}A_{se}\left(\frac{d_{se}-y}{y}\right)(d_{se}-0.04y)$$
(12)$$P_{u}=\frac{2M_{u}}{L_{a}}$$

### 4.2. Prediction Spacing and Width of Cracks

The prediction of flexural crack spacing and width of the control and strengthened beams calculated using equations as stipulated in the Euro-code 2 [22] was founded on the modular ratio of the main steel rebars and CFRP fabrics and the position of the neutral axis (N.A.) of the strengthened beam cross-section. The crack spacing and width of the E-SBR CFRP fabrics strengthened beams are calculated using Equations (13)–(19).
(13)$$S_{m}=3.4c+0.425k_{1}k_{2}\frac{D_{b}}{\rho_{eff}}$$
(14)$$\rho_{eff}=\frac{A_{s}+n_{se}A_{se}}{A_{ceff}}$$
(15)$$A_{ceff}=\min\left\{\begin{array}{l} 2.5\times b\times c\\ b\times (h-y)/3 \end{array}\right\}$$
(16)$$n_{se}=\frac{E_{se}}{E_{c}}$$
(17)$$w_{k}=S_{m}\left(\varepsilon_{sm}-\varepsilon_{cm}\right)$$
(18)$$\varepsilon_{sm}-\varepsilon_{cm}=\frac{\sigma_{s}-k_{t}\frac{f_{ct}}{\rho_{eff}}(1+\alpha_{e}\rho_{eff})}{E_{s}}\geq 0.6\frac{\sigma_{s}}{E_{s}}$$
(19)$$\alpha_{e}=\frac{E_{s}}{E_{c}}$$

### 4.3. Prediction of Deflection

The load versus deflection graph of E-SBR CFRP fabrics strengthened RC beams can be divided into three different linear phases [23]. These three phases of the curve are plotted in Figure 5. However, the model does not conceive the influence of delamination of CFRP fabrics in the E-SBR technique on the prediction of the deflection curve.

#### 4.3.1. Pre-Cracking Phase

In this phase, the deflection of the E-SBR CFRP strengthened specimens is calculated based on elastic beam deflection as the cracks have not yet been initiated nor propagated. The effective moment of inertia (*I_e_*) of the section is assumed to be equivalent to the gross moment of inertia (*I_g_*) of the section. The deflection is calculated based on Equation (20).
(20)$$\Delta =\frac{PL_{a}(3L^{2}-4L_{a}{}^{2})}{48E_{c}I_{g}}$$

#### 4.3.2. Cracking Phase

In the cracking phase, the section of the beam does not comprise a constant moment of inertia. Thus, an effective moment of inertia (*I_e_*) of the section is used for the calculation of deflection. The effective moment of inertia (*I_e_*) is greater than the cracked moment of inertia (*I_cr_*) and less than the gross moment of inertia (*I_g_*), which depends on the crack’s propagation, load, and tensile resistance of the concrete. The most accepted equation for estimating the effective moment of inertia (*I_e_*) was developed by Branson [24] and is given in Equation (21). In this phase, the deflection is evaluated using the Equation (22).
(21)$$I_{e}=I_{cr}\left[1+{\left(1-\frac{M}{M_{y}}\right)}^{3}\right]$$
(22)$$\Delta =\frac{PL_{a}(3L^{2}-4L_{a}{}^{2})}{48E_{c}I_{e}}$$

#### 4.3.3. Post-Cracking Phase

In the post-cracking phase, it is complicated to calculate the deflection due to the nonlinear stress-strain relationship in the concrete. The deflection of the beam may be calculated by integrating the curvature along the span length of the beam; however, the determination of the deflection in this process can be time-consuming and tedious. Hence, the deflection can be determined by simplifying the moment–curvature equation and assumed the bilinear relationship of the E-SBR CFRP strengthened RC beams. Therefore, the evaluation of curvature after the yielding of steel rebar obtained applying linear interpolation between *ϕ_y_* (yield curvature) and *ϕ_u_* (ultimate curvature), which is shown in Equation (23). An effective moment of inertia (*I_e_*) and deflection of the specimens can be calculated by Equations (22) and (24), respectively.
(23)$$\phi =\phi_{y}+\frac{\left(M-M_{y}\right)}{M_{u}-M_{y}}(\phi_{u}-\phi_{y})$$
(24)$$I_{e}=\frac{M}{\phi E_{c}}$$

## 5. Results and Discussions

### 5.1. Flexural Capacity

The first cracking load is vital as the flexural stiffness reduces due to the formation of the first crack [25,26]. In this research work, it was found that the first cracking load increased up to 141 and 118% for the series of strengthened specimens A and B, respectively, compared to the control beam as revealed in Figure 6a. This could be attributed to the enhanced pre-cracking stiffness by the side-EBR technique with CFRP fabrics.

The serviceability limit states comprise of reliable predictions of the prompt and long-term deformation of the structures [27]. To fulfill the serviceability limit states, the reinforced concrete structure must be serviceable and accomplish its intended task during its functioning life. The service load of the structural element is defined as 60% of its ultimate load [28]. The service load enhanced up to 2.73 and 2.39 times for the series of strengthened specimens A and B, respectively, compared to the control beam. By contrast, Almusallam et al. [29] reported that the service loads of strengthened RC beams using four and two layers of GFRP and CFRP laminates by EBR technique, were increased by about 1.65 and 1.77 times, respectively.

The flexural performance of the strengthened beams compared to the control beam is exposed in Figure 6b. The RC beams strengthened with side-EBR technique utilizing CFRP fabrics significantly increased the flexural performance. The beams with E-SBR CFRP fabrics and end anchorage increased the flexural load capacity up to 174 and 140% correspondingly, compared to the control beam. Thus, the higher flexural load-bearing capacities of the strengthened beams of this research could be compared to the beams strengthened with existing EBR system using 1.40 and 4.78 mm thickness of CFRP sheets, which shown the ultimate load increment of 42 and 58%, respectively [30].

### 5.2. Load-Deflection Responses

The load versus deflection graphs for the reference (CB) and E-SBR strengthened beams are exhibited in Figure 7 (series-A and series-B with different main reinforcement ratio). The characteristics of the strengthened beams are enhanced compared with the un-strengthened beam (CB). However, the new strengthening E-SBR technique delays the opening and growth of cracks of the strengthened beams. The first crack of the strengthened beams is initiated at higher load with lower deflection. The E-SBR strengthened beams show three distinct stages of load–deflection behavior. The first stage comprises of the almost linear load–deflection curves, which represent uncracked beams due to the high bending stiffness of the specimens. In the second stage, cracks formation and propagation have reduced the stiffness of the specimens. However, the linear characteristics of the specimens were also perceived. The cracking load of the control beams is 10.50 and 12.60 kN, whereas the strengthened beams cracking load varied from 15.10 to 25.30 kN and 16 to 27.50 kN for series A and B, respectively (Figure 6a). The third stage of the load–deflection curve was initiated owing to the yielding of the steel reinforcement. The yielding loads of steel for the two control beams were found as 30 and 35 kN, whereas for the strengthened beams, these loads ranged from 45 to 65 kN and 50 to 70 kN for series A and B, respectively. It has been noticed that the bending stiffness of all beams reduced at this stage. The rate of reduction of the stiffness depends on the concrete cracking due to yielding of both the main reinforcement and the strengthening fabrics. The rate of reduction of the stiffness of the strengthened beam was found lower compared with the CB and this could be due to the contribution of the side strips of CFRP fabrics that enhanced the flexural performance.

### 5.3. Failure Modes

The failure mechanisms of the control and strengthened beams are displayed in Figure 8. The failure of the control beams CB-A and CB-B occurred after significant tensile strain developed in the longitudinal main reinforcement and plastic hinge appeared in the central moment region in a ductile manner and this mode of failure is consistent with the design approach.

In the beams (SE-A and SE-B) strengthened with 42 mm width of CFRP fabrics, the failure was governed by fabrics rupture in between two loading point as demonstrated in Figure 8b. However, the 84 and 125 mm width of CFRP fabrics used strengthened beams failure mode was delamination of fabrics from the concrete substrate. The CFRP fabrics delamination occurred owing to the flexural cracks opened in the consistent moment zone of the beams as the applied load increased; this initiated debonding among the CFRP fabrics and the concrete substrate and the failure spread in the direction of the shear span until the end of CFRP fabrics separated from the concrete surface. The bond strength is not sufficient between the fabrics and concrete substrate to confirm the rupture of the fabrics when the width of the fabrics more than 42 mm. It is found that if the E-SBR width of strips more than 42 mm, the mode of failure the strengthened beams are controlled by the bond strength between the CFRP fabrics and the concrete.

Conversely, the lower bond width of the CFRP fabrics of 42 mm, the beams reached its ultimate capacity and failed due to composite action [14]. In order to eliminate the delamination failure of the beams SE-3A and SE-3B were strengthened with 100 mm width U-wrap end anchorage. However, the U-wrap width was not enough to prevent the delamination or peeling off failure. Therefore, the debonding of E-SBR strengthened beams could be eliminated by increasing the width of U-wrap end-anchor or applying the mechanical anchorage system. The debonding failure of the EBR-FRP sheets strengthened beams can be postponed by the externally bonded reinforcement into the grooves technique [31].

### 5.4. Cracking Behavior

In this study, the crack width of the beams was enumerated utilizing Dino-Lite (TMS-AM73115MTF) digital microscope at the load increment of 5 kN. The correlation between the applied load and crack width of the control and strengthened beam specimens are exhibited in Figure 9. As known, the cracks occur in the beam specimens as the concrete tensile stress exceeds the limiting tensile strength of concrete. Initially, the stress in the concrete progressively developed via the bond strength among the concrete and the reinforcements. Once the stresses in the concrete increases and over the tensile strength of concrete, then new cracks form [32]. As expected, the first flexural crack was initiated at the constant bending regions of the beam specimens. The load–crack width graphs (Figure 9) expressed a linear increment of crack width until the yielding of the reinforcement and after that, the width of crack increased rapidly. However, the strengthened specimens exhibited a lesser width of cracks. Table 3 shows the number and the mean spacing of cracks of the beams.

Therefore, the E-SBR technique with CFRP fabrics has significantly increased the number of cracks and thus decreased the average spacing of the cracks. Hence, this technique has a positive influence on controlling the width of cracks.

### 5.5. Ductility Analysis

Ductility is one of the strategic design parameters for structural elements. The utmost significant aspect of ductility is to provide advance caution before failure of the structures [32]. The ductility and deformability of RC structures are usually measured by the ductility and deformability indices [33]. The indices are defined as the following expression:(25)μΔu=ΔuΔy
(26)μΔf=ΔfΔy

Where, $\mu_{\Delta u}$ and $\mu_{\Delta f}$ are the ductility and deformability indices at ultimate and failure load, respectively,$\Delta_{y}$, $\Delta_{u}$, and $\Delta_{f}$ are the deflection at tension steel yield, ultimate and failure load, correspondingly. The indices of the beam specimens and comparison between the strengthened and control beams are revealed in Table 4.

The ductility index for series A and B strengthened specimens decreased up to 2.28 and 2.62, respectively, and deformability reduced up to 2.53 and 2.92, respectively. This result concluded that the ductility and deformability indices reduce due to increasing in the strengthening reinforcement and debonding failure mode. The ductility and deformability indices of the strengthened RC beams decreased with the increase in the strengthening CFRP reinforcement [34].

### 5.6. Energy Absorption Capability

Before failure of the RC beams, the specimens are expected to absorb the energy. This characteristic is very crucial for the design of structures considering the aspects of the earthquake. The energy absorption capability of the beam specimens is assessed by the total area underneath the load versus deflection graph [35,36]. Figure 10 graphically represents the energy absorption capacities of the control and strengthened beams. The E-SBR strengthening technique with CFRP fabrics successively increased the energy absorption capability compared to the control beams. The first and second series of the beams exhibited an enhancement of energy absorption capability of 49 and 58%, respectively, due to enhancing the flexural capacities of the beams strengthened with of CFRP fabrics.

By contrast, the RC beams strengthened with EBR technique utilizing steel and CFRP plate reduced the energy absorption capability of 58 and 36%, respectively [37]. The flexurally strengthened RC beams by existing NSM method with CFRP bars had decreased the energy absorption capability up to 49% [38]. The NSM-CFRP strips strengthened RC beams revealed a reduction in the energy absorption capability up to 38% [39].

### 5.7. Stiffness

Stiffness is one of the essential characteristics of the RC structural members at serviceability stage. Enhance stiffness of RC structures has a significant influence on the essential properties, such as crack and deflection. The stiffness of the control and strengthened beams was evaluated based on the gradient of the load versus deflection curves as shown in Figure 7. The stiffness of the specimens was found to be linear until the first cracking load and all the specimens exhibited the reduction in the stiffness after the first cracking load. As known, the RC beams carry the service loads in the pre-cracking phase [40] and hence, it is very crucial to assess the enhancement of stiffness owing to the strengthening of the beams at the service load level [33]. The service load limit is determined as the applied load from the first cracking load to the load obtained based on the point of deflection, which is equal to the span length of the specimen/480 [41]. In this study, the deflection of the beam at the service load was found as 4.17 mm. The stiffness of the beams is graphically displayed in Figure 11. The E-SBR technique with CFRP fabrics controlled the initiation and spreading of the cracks, which in turn led to an increase in the overall stiffness of the strengthened beams. Existing research specified that the widths and average spacing of cracks are lower in the strengthened RC beam owing to superior stiffness [42,43]. The E-SBR technique with CFRP fabrics used in this research improved the stiffness from 29 to 91% and 15 to 72% for series A and B, respectively. Further, the beams strengthened with E-SBR and end anchorage using CFRP fabrics significantly enhanced the stiffness of the specimens. Therefore, the E-SBR technique utilizing the CFRP fabrics is very competent for enhancing the stiffness of the beams, due to the redistribution of stresses.

### 5.8. Effectiveness of E-SBR Technique

The parameters, such as deflection, concrete compressive, and steel rebar tensile strains have been reduced owing to the strengthening of RC beams applying E-SBR method with CFRP fabrics, as presented in Figure 10. The decrease of deflection of the strengthened beams by a maximum of about 58, 46, and 51% for series A and 56, 54, and 41% for series B at applied loads of 10, 20, and 30 kN, correspondingly, compared to the control beam (Figure 12a). The concrete compressive strains of the strengthened beams have been reduced by about 42, 28, and 29% for series A and 43, 27, and 21% for series B at equivalent applied loads of 10, 20, and 30 kN, correspondingly compared with the reference beam (Figure 12b). Correspondingly, the main steel rebars tensile strain of the strengthened beams also decreased by about 85, 54, and 60% for series A and 82, 65, and 58% for series B at equivalent employed loads of 10, 20, and 30 kN, correspondingly, compared with the reference beam (Figure 12c). This lessening could be attributed to the enhanced flexural stiffness of the strengthened beams by the E-SBR technique using CFRP fabrics.

## 6. Validation of Analytical Models

The analytically predicted outcomes and the experimental results of the control and strengthened beams are shown in this section.

### 6.1. Validation of Load

The relationship among the experimental and predicted loads at yield and ultimate stages of the control and strengthened RC beams is graphically exhibited in Figure 13 and Figure 14, respectively. The prediction model is introduced to be very effective for predicting the yield and ultimate load carrying capability of beams in flexure as these values are closer to the experimental values.

### 6.2. Validation Spacing and Width of Cracks

The predicted and experimental flexural crack spacing of the beams is distinctly shown in Figure 15. The predicted and experimental values of the beams have a very good agreement. It is perceived that all the strengthened beams have smaller crack spacing than the control beam. On the contrary to the crack spacing, the predicted, and experimental crack widths of the beam as revealed in Figure 16 do not have a close agreement. This could be attributed to the effect of the E-SBR technique that increases the number of cracks, which in turn, decreases the crack width of the RC beams.

### 6.3. Verification of Deflection

The interrelation of the predicted and experimental load versus mid-span deflection of the beams is presented in Figure 17. The relationship between the predicted and experimental results of the tested beams is in close agreement except for the beam specimens that suffered delamination failure. Since the model was not capable to predict the deflection after yielding the reinforcements due to the damage formed by the higher stress transfer to the E-SBR CFRP fabrics.

## 7. Conclusions

This study has focused on the flexural behavior of RC beams strengthened with newly proposed E-SBR technique using CFRP fabrics through the epoxy resin. The flexural enhancement of RC strengthened beams was compared with the control beam. In addition, the efficiency of the new E-SBR technique was studied. The following conclusions were made from the experimental and analytical investigation:The E-SBR technique using CFRP fabrics efficiently enhanced the first cracking and ultimate load up to 141 and 174%, correspondingly, compared to the control beam. Hence, the E-SBR technique might be potential for implementation in practice compared with the existing EBR technique.New E-SBR CFRP fabrics significantly enriched the service load up to 2.73 times compared with the control beam by reducing deflection at different load levels.Proposed E-SBR technique with 42 mm width CFRP fabrics strip strengthened RC specimens exhibited rupture, while the beams with 84 mm width of fabrics failed by concrete cover separation; however, beams strengthened with 125 mm width of fabrics with U-anchor failed by delamination of fabrics. Thus, the ductility and deformability of those strengthened specimens reduced up to 38% compared to the control beam.The E-SBR strengthening technique with CFRP fabrics improved the stiffness and energy absorption capabilities up to 91 and 58%, correspondingly, compared with the control specimen.The analytical model predicts the load, deflection, and cracks spacing of the strengthened E-SBR-CFRP specimens that are in adjacent agreement to the experimental results.Therefore, the proposed E-SBR technique is appropriate when structures facing spalling of concrete or difficulty of implementation of existing EBR technique due to no space beneath the beams or required to attain the additional flexural capacity without disturbing the structures because of the flexibility of the technique to increase the bonding area of strengthening reinforcement.

## Figures and Tables

**Figure 1 materials-14-02809-f001:**
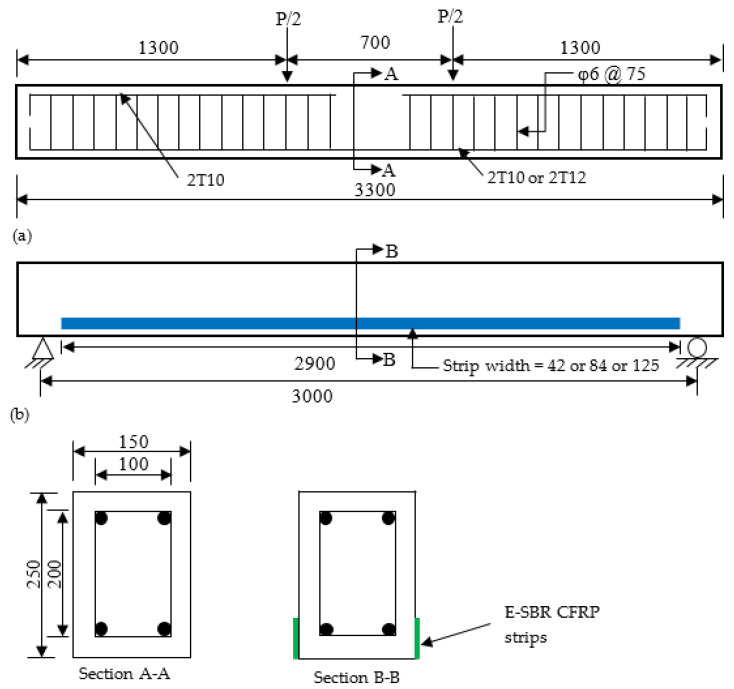
Detailing of the beam specimens (dimensions are in mm). (**a**) control beam; (**b**) strengthened beam.

**Figure 2 materials-14-02809-f002:**
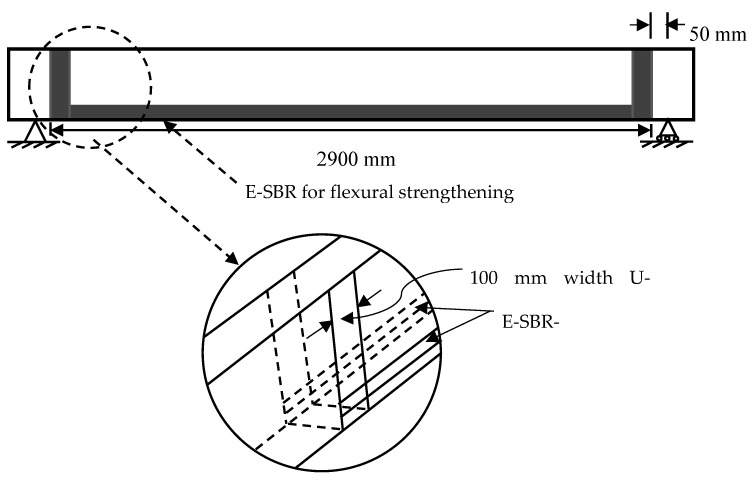
U-wrap end anchorage.

**Figure 3 materials-14-02809-f003:**
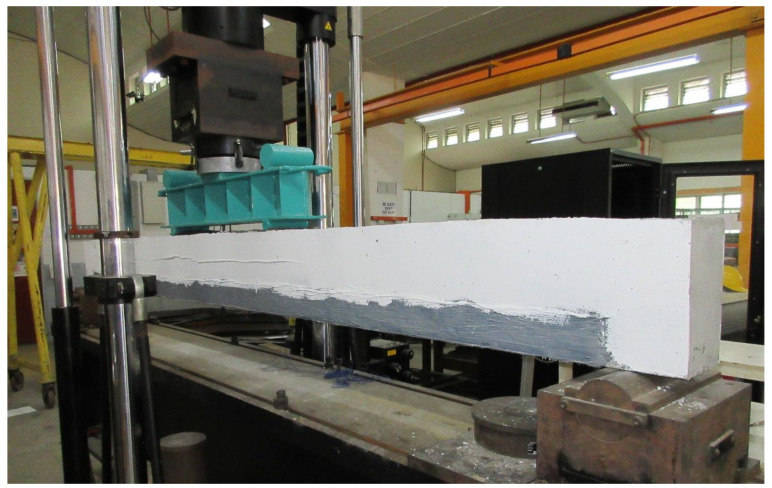
Experimental setup.

**Figure 4 materials-14-02809-f004:**
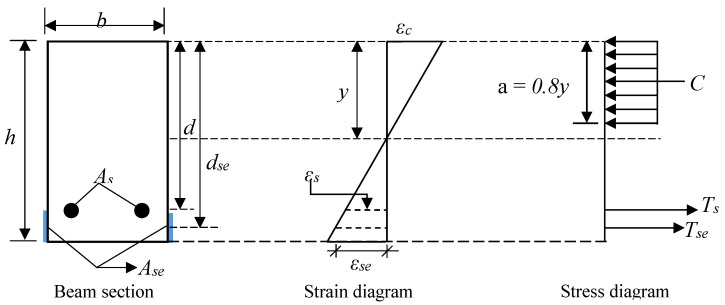
Section of E-SBR CFRP strengthened beams.

**Figure 5 materials-14-02809-f005:**
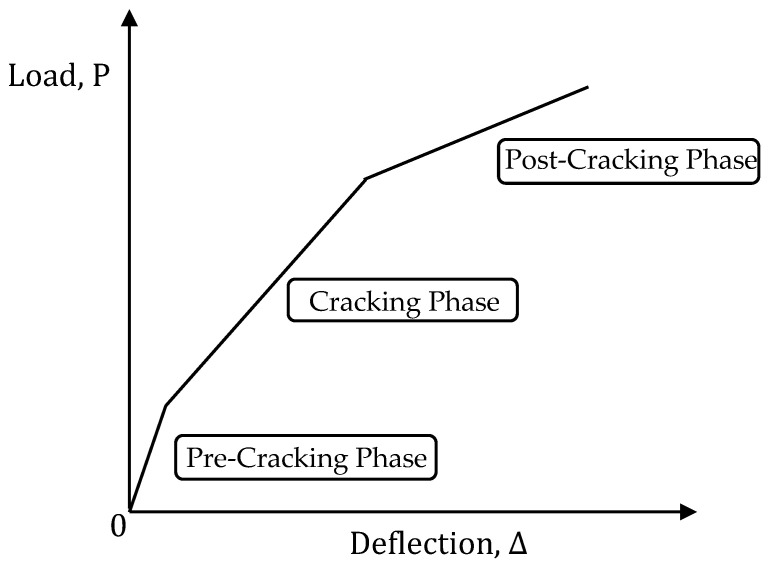
Schematic model of load-deflection graph.

**Figure 6 materials-14-02809-f006:**
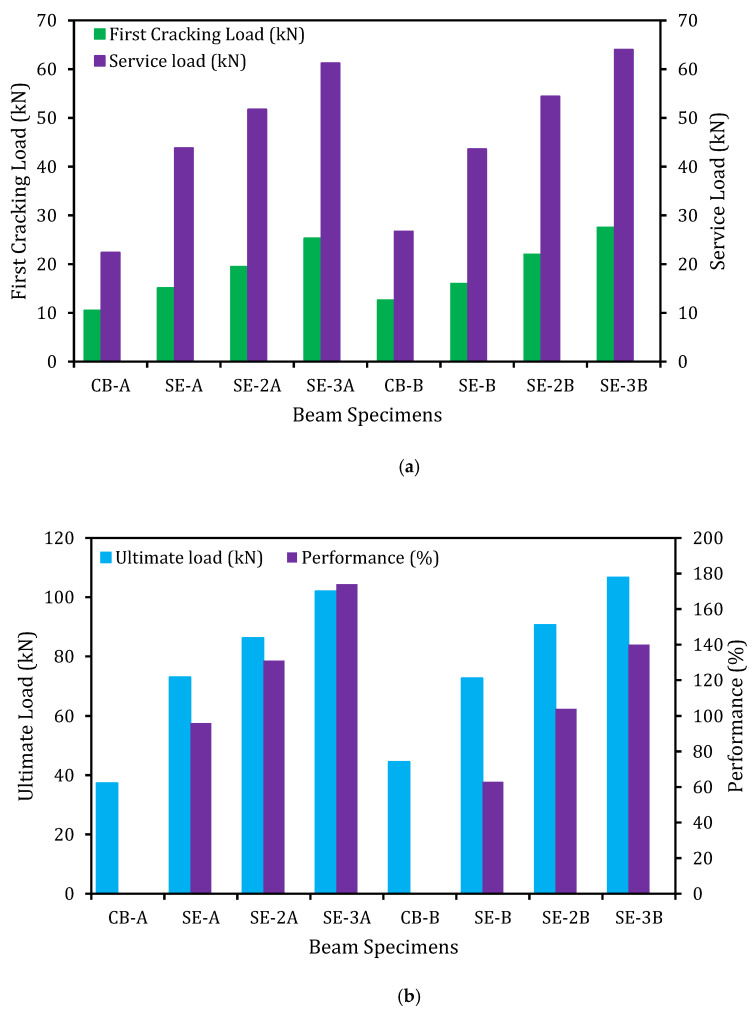
Flexural loads improved by E-SBR technique. (**a**) First cracking and service load. (**b**) Ultimate load.

**Figure 7 materials-14-02809-f007:**
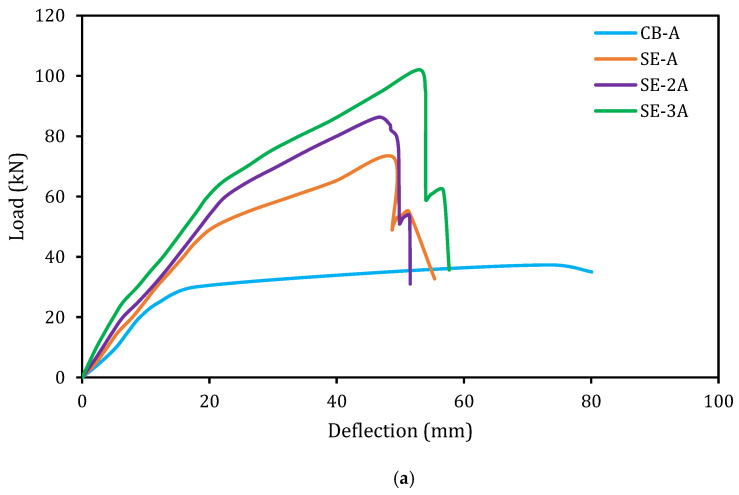
Load–deflection graphs. (**a**) series-A; (**b**) series-B.

**Figure 8 materials-14-02809-f008:**
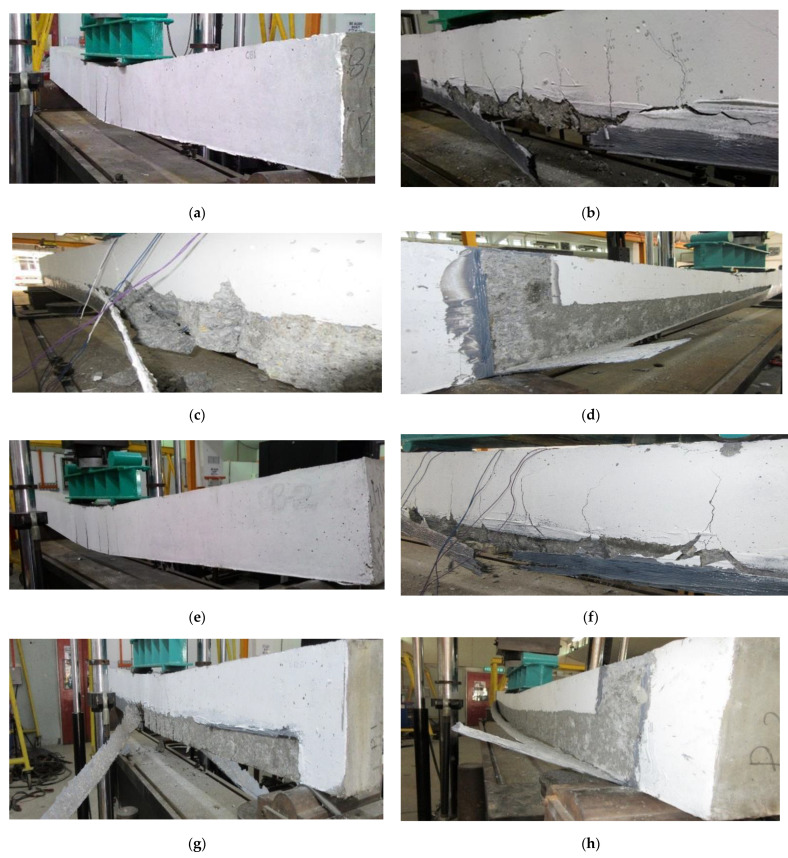
Failure modes of the beams; (**a**) CB-A; (**b**) SE-A; (**c**) SE-2A; (**d**) SE-3A; (**e**) CB-B; (**f**) SE-B; (**g**) SE-2B; (**h**) SE-3B.

**Figure 9 materials-14-02809-f009:**
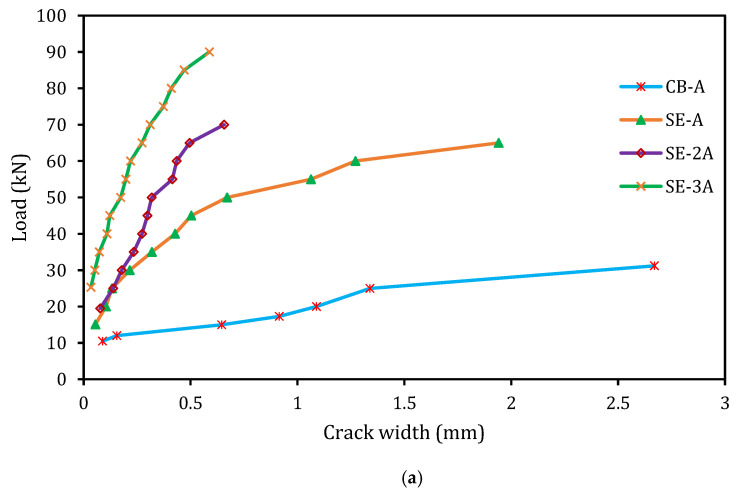
Crack width at several load level. (**a**) series-A; (**b**) series-B.

**Figure 10 materials-14-02809-f010:**
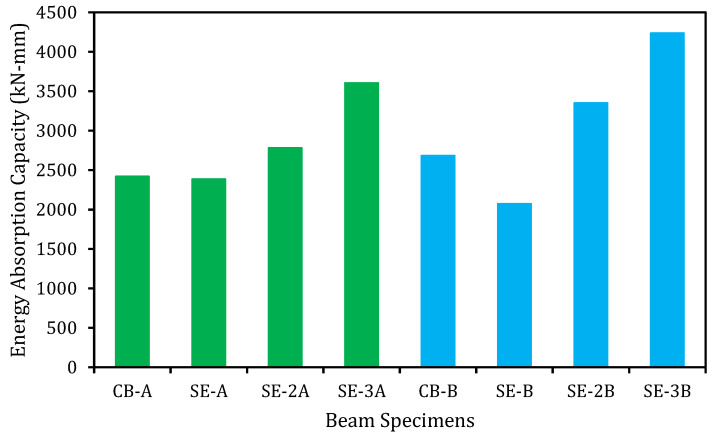
Energy absorption capacity of side-EBR strengthened beams.

**Figure 11 materials-14-02809-f011:**
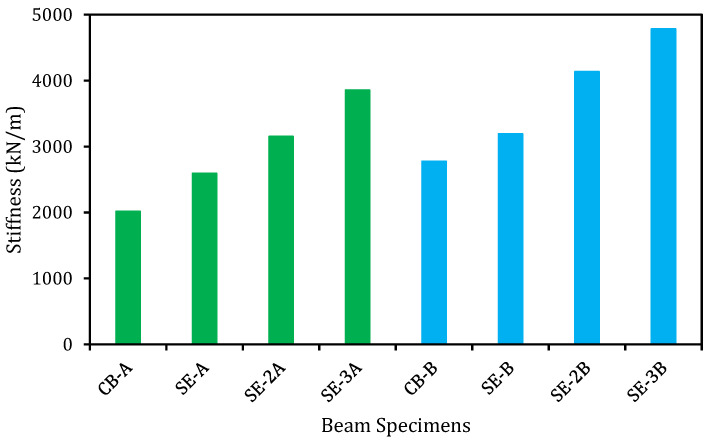
Stiffness of E-SBR strengthened beams.

**Figure 12 materials-14-02809-f012:**
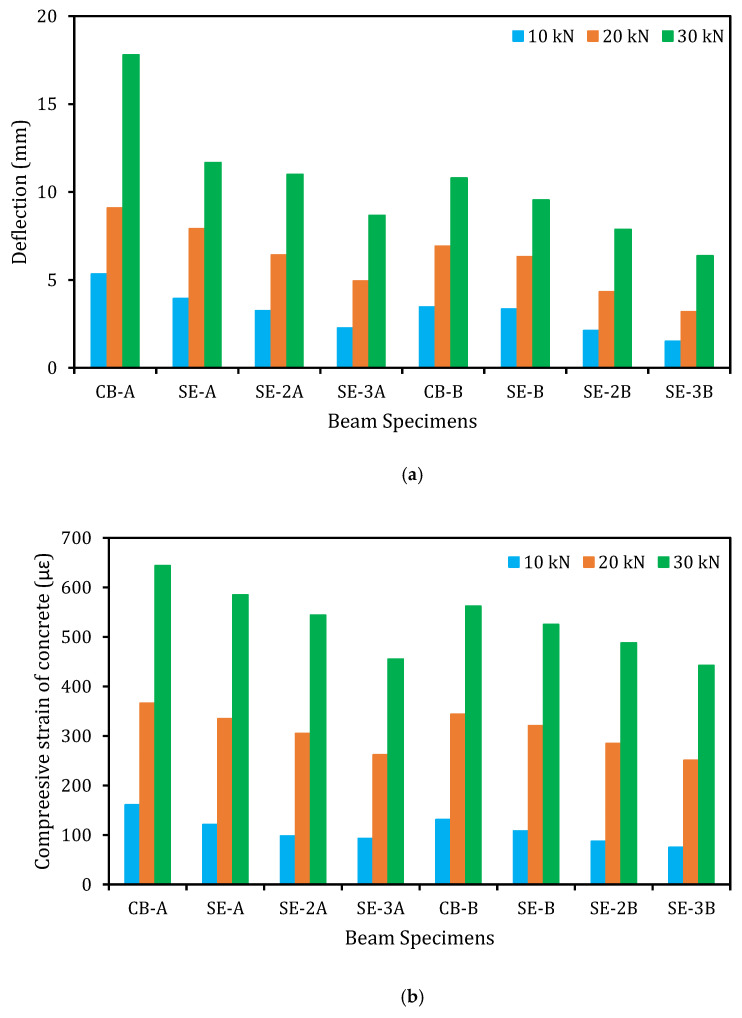
Reduced deflection, compressive, and tensile strains by E-SBR-CFRP fabrics at different loading. (**a**) reduced deflection; (**b**) reduced concrete compressive strains; (**c**) reduced steel rebar tensile strains.

**Figure 13 materials-14-02809-f013:**
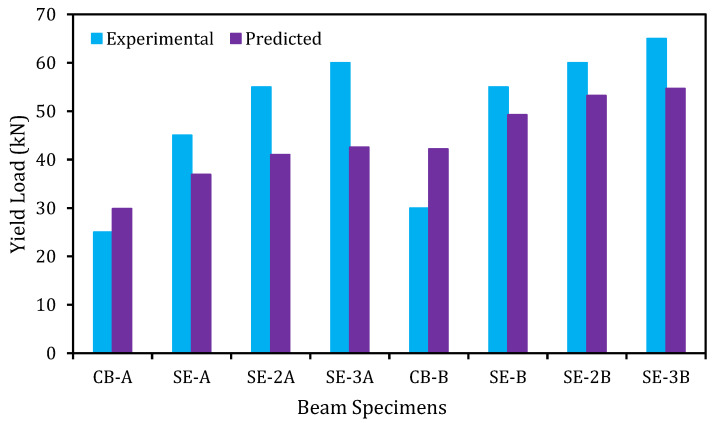
Experimental and predicted yield load comparison.

**Figure 14 materials-14-02809-f014:**
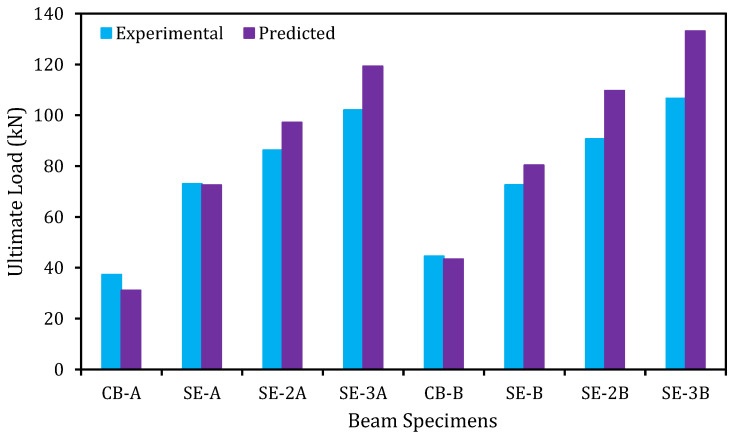
Experimental and predicted ultimate load comparison.

**Figure 15 materials-14-02809-f015:**
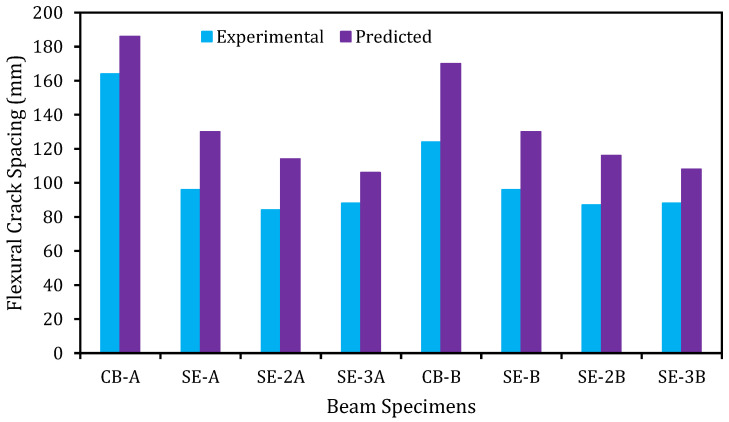
Experimental and predicted crack spacing.

**Figure 16 materials-14-02809-f016:**
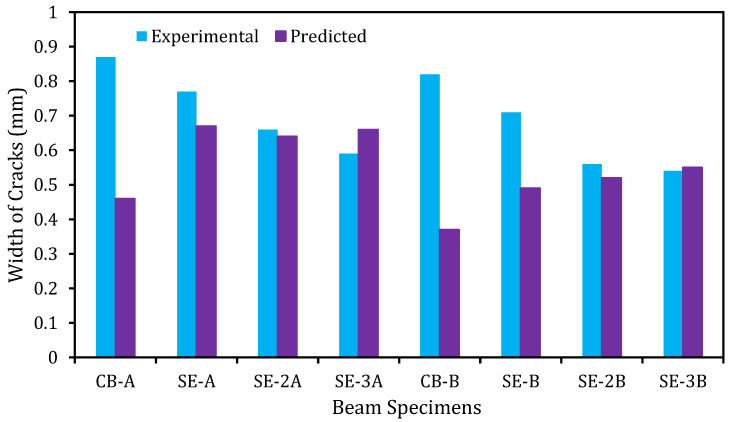
Experimental and predicted crack width comparison.

**Figure 17 materials-14-02809-f017:**
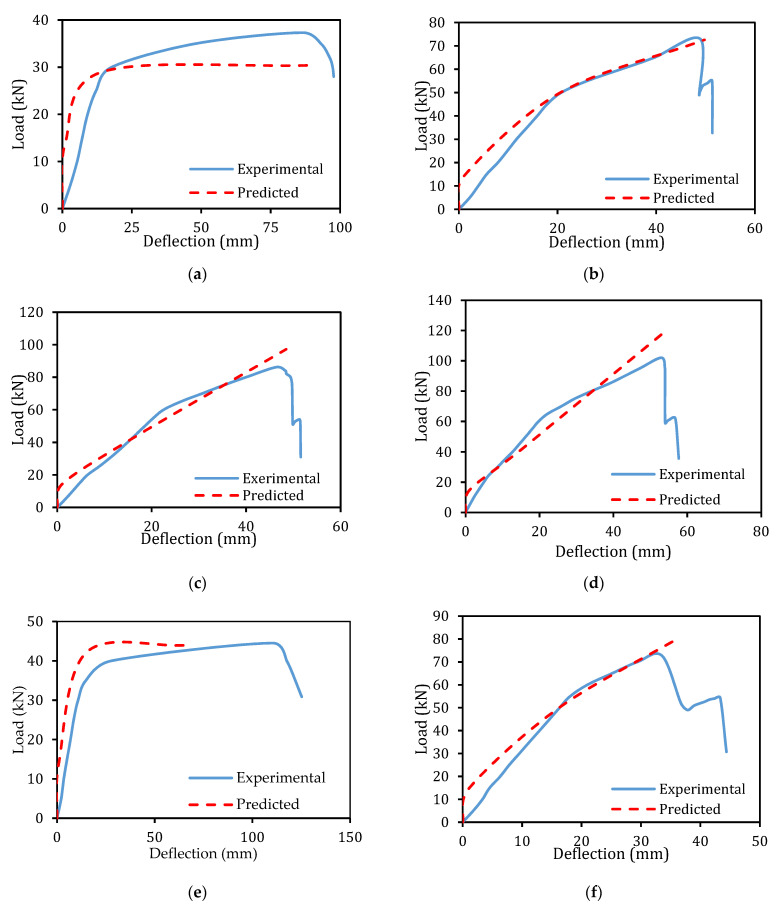
Predicted load-deflection graph comparison. (**a**) CB-A; (**b**) SE-A; (**c**) SE-2A; (**d**) SE-3A; (**e**) CB-B; (**f**) SE-B; (**g**) SE-2B; (**h**) SE-3B.

**Table 1 materials-14-02809-t001:** Test matrix.

Specimen Designation	Ratio of Reinforcing Steel	Strengthening Configurations			
		Materials	No. of Layers	Width of Strips (mm)	U-Wrap Anchor
CB-A	0.005	-			
SE-A		CFRP Sheet	2	42	-
SE-2A				84	-
SE-3A				125	100 mm width
CB-B	0.0071	-			
SE-B		CFRP Sheet	2	42	-
SE-2B				84	-
SE-3B				125	100 mm width

**Table 2 materials-14-02809-t002:** Mix proportion of concrete.

W/C Ratio	Quantity (kg/m^3^)			
	Water	Cement	CA	FA
0.46	210	457	948	745

**Table 3 materials-14-02809-t003:** Tested results for cracks.

Specimen Designation	Properties of the Cracks	
	Number	Mean Spacing (mm)
CB-A	13	164
SE-A	20	96
SE-2A	24	84
SE-3A	23	88
CB-B	16	124
SE-B	20	96
SE-2B	24	87
SE-3B	24	88

**Table 4 materials-14-02809-t004:** Summary of indices.

Beams ID	Ductility Index	Ratio to CB	Deformability Index	Ratio to CB
CB-A	6.06	1.00	6.59	1.00
SE-A	2.73	0.45	3.09	0.47
SE-2A	2.28	0.38	2.53	0.38
SE-3A	2.68	0.44	2.92	0.44
CB-B	6.34	1.00	6.82	1.00
SE-B	2.80	0.44	3.23	0.47
SE-2B	2.62	0.41	2.92	0.43
SE-3B	2.71	0.43	3.47	0.51

## Data Availability

Data sharing is not applicable to this article.

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
