# Peer review of "Flexural Performance of RC Beams Strengthened with Externally-Side Bonded Reinforcement (E-SBR) Technique Using CFRP Composites"

_materials, 2021, doi:10.3390/ma14112809_

Round 1

Reviewer 1 Report

I think it is a good idea to use CFRP to supplement the strength of concrete beams. Overall, I think the appropriate results for the reinforcing effect were also expressed.

However, there are a few things to check.

1. There will be an issue with adhesives during the side reinforcement process. It is judged that the strength of CFRP has not been fully transmitted and destruction has progressed. I think that it is possible to design future improvement directions only when the limit strength can be evaluated.

2. The process of bonding to the side with an adhesive is practically a difficult process for a number of reasons. I think there will be problems such as price, processing time, and product uniformity. From this point of view, is there any significance of future research projects?

3. Wouldn't the inner reinforced structure be more advantageous than the side bonding? CFRP has very good mechanical strength. Can side bonding provide a proper reinforcing effect? Readers may decide that the author has used excessive material. 

Reviewer 2 Report

The present paper introduces an experimental approach for flexural strengthening of RC beams with external side Bonded Reinforcement.

They demonstrated that TC beams exhibited superior energy absorption capacity, stiffness and ductile response.

The paper is well written, and the authors demonstrate knowledge of the topic.

The figure and the table are clear.

In my opinion, the paper deserves to be published in this journal after a major revision procedure needed to improve the research soundness of the article.

Comments and Recommendations:

1. At the end of the paper, you should put the outline of the paper in order to make the article more readable.

2. The nomenclature Section should stay at the beginning.

3. The literature review is lacking some more interesting aspects that you should briefly discussed:

-Numerical simulations of the delamination: see the following papers and others 10.3390/FIB8060042 , 10.1177/1099636218824873

4. The results of your curves look smoothed. Did you use a smoothing algorithm?

5. Section 6.3 is potentially the most interesting of the paper, but it lacks in several parts. The discussion in terms of the curve is not enough. I suggest improving it.

6. The are some misspelling words in the paper. Please read carefully.

7. The conclusion section is well written.

Reviewer 3 Report

Dear Authors,

Thank you for your manuscript, here will be few of the comments:

Line13: please insert abbreviation RC at first mentioning.

The Introduction requires a bit of more elaboration and references. Also, there is missing clear identification of the goal and objectives of your paper. Please clearly indicate the novelty of your method and bring some references to the methods which exist so far and indicating those disadvantages.

You restructure a bit your manuscript, for example, please indicate clearly chapter Materials and Methods where you describe all materials, methods and analytical approaches. Then make chapter Results and Discussions. Once again check formatting for the references.

In order your suggested technique could potential for implementation can you please indicate clearly in your conclusions that there were no attempts to smth similar by other researchers previously and what was the lack of the previous research. If you mention once in the text ‘new or novel’ please use this term also in your conclusions.

Round 2

Reviewer 2 Report

The paper can be accepted in the present form